# Integration of extreme risk protection orders into the clinical workflow: Qualitative comparison of clinician perspectives

Kelsey M. Conrick[1,2]*, Sarah F. Porter[1], Emma Gause[2], Laura Prater[2], Ali Rowhani-Rahbar[3], Frederick P. Rivara[2,3,4], Megan Moore[1,5]

1 School of Social Work, University of Washington, Seattle, Washington, United States of America, 2 Firearm Injury & Policy Research Program, Seattle, Washington, United States of America, 3 Department of Epidemiology, School of Public Health, University of Washington, Seattle, Washington, United States of America, 4 Department of Pediatrics, School of Medicine, University of Washington, Seattle, Washington, United States of America, 5 Harborview Injury Prevention & Research Center, Seattle, Washington, United States of America

* kmc621@uw.edu

**Data Availability Statement:** All relevant data are within the paper and its Supporting information files.

## Abstract

Extreme risk protection orders (ERPO) seek to temporarily reduce access to firearms for individuals at imminent risk of harming themselves and/or others. Clinicians, including physicians, nurse practitioners, and social workers regularly assess circumstances related to patients' risk of firearm-related harm in the context of providing routine and acute clinical care. While clinicians cannot independently file ERPOs in most states, they can counsel patients or contact law enforcement about filing ERPOs. This study sought to understand clinicians' perspectives about integrating ERPO counseling and contacting law enforcement about ERPOs into their clinical workflow. We analyzed responses to open-ended questions from an online survey distributed May-July of 2021 to all licensed physicians (n = 23,051), nurse practitioners (n = 8,049), and social workers (n = 6,910) in Washington state. Of the 4,242 survey participants, 1,126 (26.5%) responded to at least one of ten open-ended questions. Two coders conducted content analysis. Clinicians identified barriers and facilitators to integrating ERPOs into the clinical workflow; these influenced their preferences on who should counsel or contact law enforcement about ERPOs. Barriers included perceptions of professional scope, knowledge gaps, institutional barriers, perceived ERPO effectiveness and constitutionality, concern for safety (clinician and patient), and potential for damaging provider-patient therapeutic relationship. Facilitators to address these barriers included trainings and resources, dedicated time for counseling and remuneration for time spent counseling, education on voluntary removal options, and ability to refer patients to another clinician. Participants who were hesitant to be the primary clinician to counsel patients or contact law enforcement about ERPOs requested the ability to refer patients to a specialist, such as social workers or a designated ERPO specialist. Results highlight the complex perspectives across clinician types regarding the integration of ERPO counseling into the clinical workflow. We highlight areas to be addressed for clinicians to engage with ERPOs.

**Funding:** This publication was supported by the National Center For Advancing Translational Sciences of the National Institutes of Health under Award Number TL1TR002318 (to KMC). The content is solely the responsibility of the authors and does not necessarily represent the official views of the National Institutes of Health. This study was supported by funds from the State of Washington to the Firearm Injury & Policy Research Program. The funders had no role in study design, data collection and analysis, decision to publish, or preparation of the manuscript.

**Competing interests:** The authors have declared that no competing interests exist.

## Introduction

The United States (US) faces a unique and worsening burden of firearm-related injuries and deaths. In 2021, nearly 49,000 individuals died from firearm injuries [1], and an estimated 86,000 were injured [2]. Firearms were used in 26,320 suicides in the US during 2021—the highest since 1993 [3]. Lethal means restriction policies seek to prevent injury and death by reducing access to the most lethal means of harm. As firearms have about a 90% case-fatality rate for self-inflicted injury [4], several policies have been enacted seeking to reduce access to firearms for individuals acutely at risk of harming themselves and/or others. One such policy, extreme risk protection orders (ERPO), allow a petitioner to file a civil order with a judge to restrict firearm access for an individual who is deemed to be at imminent risk of harming themselves and/or others with a firearm. If granted, law enforcement officers remove currently accessible firearms and submit the respondent's name to a list indicating they may not be sold a firearm. ERPOs currently exist in 19 states and Washington D.C., although who can petition for an ERPO varies based on state policy. In all jurisdictions, law enforcement officers or another government entity (e.g., Attorney General) can file an ERPO; 14 states also allow a family or household member to petition [5].

Clinicians who serve those at highest risk of firearm injury, including physicians, advanced registered nurse practitioners (ARNP), and social workers, may play a critical role in ERPOs. Because most individuals seek care in healthcare settings (e.g., emergency department, routine primary care) in the period before a behavioral health crisis [6] clinicians have an opportunity to identify those at imminent risk of harming themselves and/or someone else with a firearm. Clinicians' assessment of access to lethal means can potentially reduce suicide risk in high-risk patients [7]; however, very few providers screen for access to firearms regularly [8].

Most clinicians have expressed willingness to independently file ERPOs; however, some clinicians remain hesitant due to lack of knowledge, concerns about time, and fears of liability [9–12]. Although the current model law published by the US Department of Justice calls for clinicians to be independent petitioners, only four states (Connecticut, Hawaii, Maryland, and New York) and Washington D.C. currently allow clinicians to file an ERPO. Despite being unable to independently file in most states, examining other ways in which clinicians may be involved is critical to understanding and improving ERPO implementation. In cases where clinicians cannot be independent petitioners, they may still have a critical role by counseling patients about ERPOs or connecting with law enforcement when an ERPO may be appropriate for a patient [13]. Additionally, filing for an ERPO might be part of a safety plan, which describes steps the patient will take when experiencing thoughts of harm, developed collaboratively between a clinician and patient [13]. Despite the potential benefits of including clinicians in the process, some have expressed hesitancy to counsel patients and their families about ERPOs or engage with law enforcement. We sought to better understand the concerns of physicians, nurse practitioners, and social workers who expressed hesitancy to be involved in the ERPO process in Washington State, where clinicians cannot independently file ERPOs. Specifically, we considered their concerns about the integration of ERPOs into their clinical workflow.

## Methods

### Design overview

We analyzed responses to ten open-ended questions from an online survey conducted May-June 2021 with clinicians in Washington state regarding their opinions of ERPOs. The University of Washington Institutional Review Board (IRB) approved all study procedures. Formal

consent was not obtained as the IRB deemed the research human subjects research that qualifies for exempt status; participants were informed of the nature of the survey, reminded that participation was voluntary and could be revoked at any time, and provided with contact information for the research team should they have questions or concerns. The research team had access to information that could identify individual participants; these data were used for recruitment purposes only and were not accessed during analysis. Our study team included expertise in firearm-related harm, extreme risk protection orders, clinical workflow for physicians and social workers, and survey and qualitative methods.

## Participants and procedures

Using a public records request, we obtained contact information for all actively licensed physicians, ARNPs, and social workers from the Washington Department of Health. We were unable to obtain contact information for psychologists or other mental health providers. The survey was distributed to 23,051 physicians, 8,049 ARNPs, and 6,910 social workers and was available May 10-June 14, 2021; we sent reminders every 8 days for 3 weeks. Clinicians were deemed ineligible if they had retired, moved out of state, or were on extended leave.

## Survey instrument

The survey asked clinicians about their familiarity with ERPOs and barriers and facilitators to their willingness to 1) counsel patients or their family members about ERPOs, 2) contact law enforcement about an ERPO for a patient, and 3) independently file an ERPO for a patient, if it were a legal option in Washington. We used Research Electronic Data Capture (REDCap) software to create and administer the survey [14]. Complete survey responses were received for 1,921 physicians, 940 ARNPs, and 1,381 social workers. Results for the quantitative components of the survey are available in other publications [9, 10]. For this study, we qualitatively analyzed responses to ten open-ended questions allowing for a more detailed explanation of their responses to questions regarding facilitators and barriers to counseling about ERPOs. Of the participants who responded to the survey, a subset of 487 physicians (25%), 228 ARNPs (24%), and 411 social workers (30%) responded to at least one of these questions.

## Data analysis

We used content analysis to identify themes and categories related to barriers and facilitators to clinicians' willingness to counsel patients/their families or contact law enforcement about ERPOs among those who expressed hesitancy or requested additional resources to do so [15]. Two coders (KMC, SP) read through 30 responses for each of the three clinician types and collaboratively developed an initial codebook. To ensure reliability of code application, the two coders double-coded an additional 30 responses for each clinician type and met to identify discrepancies, which were resolved through discussion until consensus was reached. Once the codebook had been refined, the coders divided the remaining responses and coded them using Dedoose analysis software [16]. The coders met regularly to debrief and discuss unclear participant responses. Once coding of all responses was complete, the research team met to develop themes and interpret patterns.

## Results

Participants identified barriers and corresponding facilitators to integrating ERPOs into their clinical workflow (Fig 1). Hesitation or unwillingness to engage with ERPOs for some clinicians was dependent on their practice setting and communities they served. Common

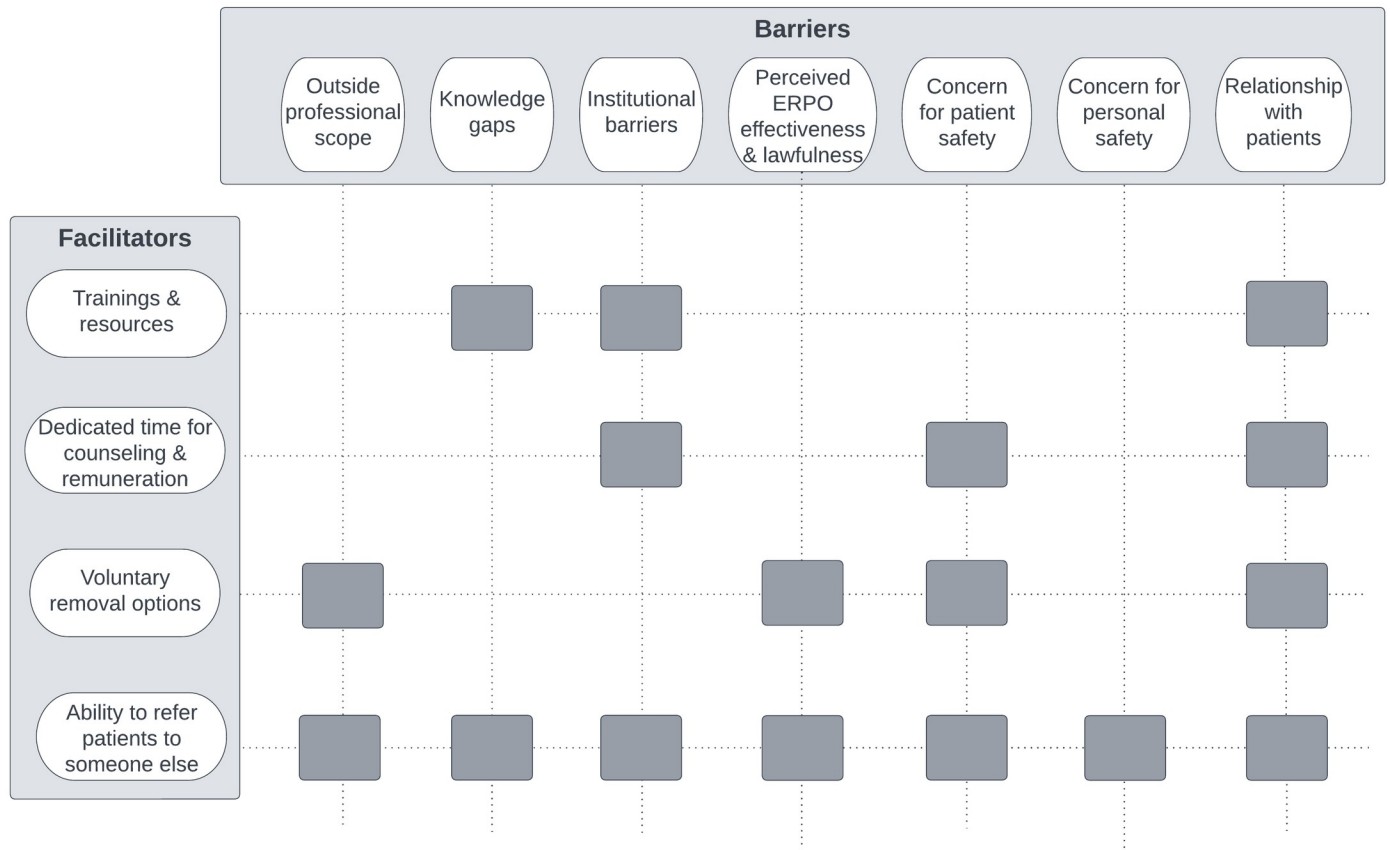

**Fig 1. Conceptual model of findings (*n* = 1,126).** The conceptual model describes which clinician-identified barriers are addressed by each facilitator to Extreme Risk Protection Order (ERPO) counseling integration into the clinical workflow. For example, trainings about ERPOs and the ability to refer patients to someone else were identified as facilitators to address the barrier of knowledge gaps.

concerns included whether ERPOs fell into their professional scope, knowledge gaps, institutional barriers, perceived ERPO effectiveness and constitutionality, and potential implications ERPOs may have on their patients, their patient's families, or themselves. Many of these barriers were particularly pressing for clinicians who cited working with patients from historically marginalized communities (e.g., Black, Indigenous, People of Color [BIPoC]). Clinicians also identified several strategies or facilitators that would address some of these barriers, including trainings and resources, dedicated time for counseling and remuneration, voluntary removal options, and the ability to refer patients to someone else. Among those who were unwilling or hesitant to engage with ERPOs in their current capacity (counsel patients/their families or contact law enforcement), these barriers and facilitators influenced their perspective of who should counsel or contact law enforcement about ERPOs. Below we discuss these themes by comparing physician, ARNP, and social worker responses, and provide representative quotes in Tables 1 and 2.

## Barriers to integrating ERPOs into the clinical workflow

**Outside of professional scope.**   Many clinicians described ERPOs as outside the scope of their profession. Some physicians and ARNPs described their professional duties as strictly medical, explaining ERPOs would fall into the scope of social workers or lawyers. A few social

**Table 1. Barriers to integration of ERPOs into the clinical workflow.** ($n$ = 1,126).

| Barrier | Representative Quote |
|---|---|
| Outside of professional scope | "Healthcare providers have a different set of tasks and priorities and have no business in counseling on purely legal matters. Majority of healthcare providers have no legal education, let alone background, to counsel on anything so far outside their profession and specialty."<br>~Physician |
| | "There has been a lot of talking about replacing police with social workers. Our jobs are not the same. As a person of color who has been mistreated by police, I do not trust the police and do not wish to work with them. I also think it is dangerous for the profession to align itself and/or be seen as an alternative to police, especially since social work is also a white-dominant field."<br>~Social Worker |
| Knowledge gaps | "As a psychiatrist in a high acuity setting I am often in the position of deciding whether or not to refer for [involuntary] detention, which will remove firearms legally. ERPO is a lower threshold and a welcome addition that I have not used for lack of knowledge."<br>~Physician |
| | "It can be hard to find information [about] many things on government websites"<br>~Social Worker |
| Perceived ERPO effectiveness and lawfulness | "This is a 'slippery slope' to go down! Where does it end? Firearms are most certainly not the only weapon a patient could use to harm themselves or others. Do we start restricting access to knives? Restrain the patient based on a 'fear' that they 'might' harm someone else or themselves?"<br>~ ARNP |
| | "Often I have clients, especially young people, say 'I don't have a gun but I can easily get one from somebody' which these tools would not capture this population. Not an easy fix, but just wanted to point this out as I hear it quite often and it always worries me as it is hard to make a safe plan around potential to obtain from other people."<br>~Social Worker |
| Relationship with patients | "I think ERPOs will be detrimental to the provider-patient relationship. No one will disclose how they are feeling if they cannot trust their providers. This is more harmful than involuntary psychiatric holds."<br>~ARNP |
| | "It would almost certainly damage any therapeutic alliance that I had developed with the patient, which could make the patient less willing or receptive to care planning with the [social worker]. Patients could also interpret this action as a violation of their trust."<br>~Social Worker |
| Concern for patient and family safety | "I worry about the safety of undocumented patients in the setting of having the authorities involved."<br>~Physician |
| | "From an [Equity, Inclusion, and Diversity] perspective, if my patient is a person of color my trust in the court system diminishes the darker their skin color is."<br>~ARNP |
| | "Transfers risk to family. I think it is my responsibility to contact police if I have concerns. If patient is willing, I would have them have family take guns away from the home, but I would measure that very closely. If patient feels threatened it could escalate very quickly."<br>~Social Worker |
| Concern for personal (clinician) safety | "I would be concerned about my own safety with initiating as there is likely lag time from when it is filed to when the patient stops having access to firearms."<br>~Physician |
| | "Some families get very angry and/or threaten to sue when we escalate care appropriately."<br>~ARNP |

(*Continued*)

**Table 1.** (Continued)

| Barrier | Representative Quote |
|---|---|
| Institutional barriers | "Very limited resources in rural areas, no social workers and deputy sheriffs are busy. Who do I send them to?"<br>~Physician |
| | "I am paid to shuffle patients through and do a lot of tests and procedures to them, not to protect them. This must change."<br>~Physician |
| | "Time is a huge factor. As it is with the responsibilities that Medical Social Workers currently have, there is not enough time in the day to get everything done. I would be very reluctant to advocate for any additional responsibilities that could necessitate court time, consultation with law enforcement or additional paperwork responsibilities to a schedule that is already overfilled."<br>~Social Worker |

workers raised concerns that ERPOs were explicitly in opposition to their profession's foundational ethics. For example, one social worker stated that because ERPOs typically involve law enforcement, they would not contact officers for an ERPO for a patient because they did not believe social work values align with policing. Additionally, social workers also described concerns about restricting patients' rights to decide when, how, or if to remove firearms from their home, explaining that their role as a provider was to support their patient's self-determination for what approach was best for them (Table 1).

**Knowledge gaps.**   Clinicians often described lack of knowledge of the referral process and patient eligibility as a barrier to the incorporation of ERPO counseling into their workflow. Social workers in particular voiced questions regarding the difference between ERPOs and other civil protection orders. Additionally, multiple clinicians disclosed that despite operating in acute mental healthcare settings for many years, this survey was the first time they had heard of ERPOs (Table 1).

**Institutional barriers.**   All three provider types cited contextual factors unique to their practice setting that created barriers or concerns to integration of ERPOs into the clinical workflow. In addition to the unique environmental considerations within and across practice settings, particularly between rural versus urban locations, common institutional barriers included staffing shortages, reimbursement structures, time, and perceived lack of institutional support for ERPOs. Clinicians disclosed that it would be unethical to provide ERPO counseling or initiate an ERPO if they were only able to see patients for a limited amount of time (e.g., 15-minute appointments, one-time welfare checks). Physicians and ARNPs reported not having staff who could support them in counseling or using ERPOs. All clinician types also described barriers regarding reimbursement structures and feeling conflicted between their need to protect patients and communities with the time they have available in their clinical workflow to complete tasks. These were often shortened due to the inability to be reimbursed for these services. Social workers sometimes questioned whether ERPO processes might be in direct conflict with policies in their practice settings. For example, some social workers explained their institutions would likely not support them to counsel patients and families about ERPOs if the patient did not consent (violating confidentiality policies). Additional workplace policies described included those that prohibit reaching out to law enforcement except when legally required (Table 1).

**Perceived ERPO effectiveness and constitutionality.**   Multiple clinicians perceived ERPOs as ineffective, and some believed such laws were unconstitutional. For example, one physician explained they did not believe their assessment of risk of engaging in a future

**Table 2. Facilitators to integration of ERPOs into the clinical workflow.** (*n* = 1,126).

| Facilitator | Barrier Facilitator Would Address | Representative Quote |
|---|---|---|
| Trainings and resources | Knowledge gaps | "We need more training and resources readily available and embedded into the [electronic health record] for primary care practitioners to be able to discuss these things with patients in an informed and evidence based way."<br>~Physician |
| | Institutional barriers | "This is the first I've heard about ERPO and firearm storage sites and I'm interested to learn more. Pamphlets for patients/family would be helpful given our time constraints for physician counseling to patients in the [Emergency Department]."<br>~Physician |
| | Relationship with patients | "Give me education and tools and I'm happy to use ERPO's. I know it will affect relationship with my client. But given the choice between our relationship and someone getting harmed, safety is the priority. I think for many of my people we could have this conversation and continue in professional relationship."<br>~Social worker |
| Dedicated time for counseling and Remuneration | Institutional barriers | "I work in urgent care and am under a lot of pressure by my employer to see more patients and faster. I would be more willing to participate in these programs if I had support from my employer."<br>~ARNP |
| | | "Being able to bill for this service. The pandemic has us running on fumes and the complexity of providing behavioral health services is already burdensome so adding yet another huge uncompensated responsibility seems like a barrier"<br>~Social worker |
| Voluntary removal options | Perceived ERPO effectiveness & lawfulness | "If a significant threat exists, then a safety plan can be put in place regarding firearms just as it has been done for decades. It is inappropriate to suspend a Constitutional right (not a privilege) for this."<br>~Physician |
| | | "It is a second amendment right. There are other actions in place such as hospitalization or asking friends and family to remove the firearms themselves."<br>~Social worker |
| | Concern for patient safety | "Most of the time, I am able to engage patients and family willingly to restrict access to lethal means through a collaborative safety plan. [. . .]. I do not have confidence in trying to engage police to file an ERPO based on my experience with working with them."<br>~Physician |
| | Relationship with patients | "Systematic biases/racism that exist in law enforcement can possibly impact physician-patient relationship especially if looking as if we are in alignment with those systems. Most of my patients at high risk are interested in [voluntarily] contracting for safety—so having voluntary do not sell list, firearm storage are more patient centered approaches with same outcomes."<br>~Physician |

(*Continued*)

**Table 2.** (Continued)

| Facilitator | Barrier Facilitator Would Address | Representative Quote |
|---|---|---|
| Ability to refer patients to someone else | Knowledge gaps | "Although I would want to be trained in all of these areas, I would hope that I can rely on a social worker who can take more ownership of these factors and is someone I can partner with to carry these issues out." ~Physician |
| | Institutional barriers | "I don't think any provider would be opposed to discussing these topics with patients. The limiting factor is time. If I could refer patients to a case management or social work who can work on the process or paperwork, I would not have an issue. Me personally filling out paperwork for this will never happen." ~ARNP |
| | Perceived ERPO effectiveness & lawfulness | "We have a constitutional right to keep and bear arms, though public and personal safety are also key. Mentally ill people already carry so much stigma I wonder if this would help or make it worse? Also, no contact orders frequently fail to prevent contact and there are other means of obtaining firearms than through legitimate sources so would an ERPO really be effective?" ~Social Worker |
| | Concern for patient safety | "I think instead of police officer involvement it should be a social worker if safe to do so. I worry about unnecessary force used against BIPOC people." ~Physician |
| | | "I think that it would be helpful to acknowledge the role of racism in historic and current determinations of danger. Giving positions of leadership to folks of color in these programs (in law enforcement, in social worker liaison roles, etc.) would personally make me feel more comfortable engaging with a process like this, particularly for BIPOC clients." ~Social Worker |
| | Concern for personal safety | "Health care personnel are at VERY high risk of on the job assault from patients and this is a threatening move. I would consider referral to social work or psych professional for this." ~ARNP |
| | Relationship with patients | "A third party system I could refer to without being directly involved, this would provide safety to the patient and protect the therapeutic relationship." ~ARNP |
| | | "I think specialized social workers (like DCR similar type of profession) could/ should do it. But not any social worker" ~Social Worker |

behavior (i.e., harming themselves and/or someone else with a firearm) justified what they deemed to be a violation of their patient's Second Amendment rights. Others explained they did not believe that ERPOs were an effective solution to preventing firearm misuse. Clinicians expressed doubts about the effectiveness of ERPOs for specific patients, citing instances where patients disclosed their ability to obtain firearms through alternative channels, despite confiscation of presently owned firearms and prohibition of purchase as a result of the ERPO.

**Concerns for patient and family safety.** Concerns over patient safety were often specific to patients' social identities, with many participants citing biases of law enforcement officers and the court system. These concerns were especially prevalent for patients who are BIPoC, undocumented, trans, or experiencing a mental or behavioral health crisis. Clinicians explained if ERPOs were being utilized to protect a patient's safety, it may be counterproductive to do so by introducing law enforcement into their household and/or community when they already experience disproportionate risk of police-related trauma, violence, and/or incarceration. Many social workers, especially those with prior negative encounters with law enforcement, emphasized that they would not utilize ERPOs in their practice setting due to the required inclusion of the legal system. Additionally, some social workers considered including a patient's family or household members in the ERPO initiation process as adding additional

risk of the patient harming them or retaliating, especially when intimate partner violence may be present (Table 1).

**Concerns for personal (clinician) safety.** Clinicians raised a variety of safety concerns, including physical safety and liability concerns. Participants described concerns that patients would physically and/or verbally retaliate against them, their families, or their office staff for referring for an ERPO. The concern of experiencing physical violence and retaliation was voiced repeatedly by providers when discussing patients who might experience mental illness. They described fears for liability for either engaging with ERPOs or failing to if they were tasked to counsel or independently file. Some clinicians were concerned that a patient or their family would sue the clinician for referring for an ERPO if the process was harmful or perceived to be an inappropriate course of action (Table 1).

**Relationship with patients.** Participants expressed concerns related to the ways in which discussing ERPOs with patients, their families, or involving law enforcement would damage the therapeutic relationship or discourage them from seeking care. Clinicians in practice settings that provide ongoing treatments not specific to behavioral health (e.g., surgeon) disclosed not wanting to jeopardize their patients' willingness to adhere to critical treatment plans by alienating them with discussions about firearms.

### Facilitators for implementing ERPOs

**Trainings and resources.** Trainings about ERPOs were identified as a facilitator to address knowledge gaps, institutional barriers, and relationship with patients. While many clinicians expressed enthusiasm at the opportunity to complete such a training, others expressed concerns that this training would be mandatory and indicated that they did not have the time or interest to complete it given other demands on their time. Clinicians also suggested that having ready-made resources to give to patients or families would address time barriers to counseling during appointments. Among social workers who expressed the significant need/ interest for additional training, some noted that a deeper understanding of ERPOs and their potential risks could be used to help educate patients and their families of their rights in the ERPO process (Table 2).

**Dedicated time for counseling and remuneration.** Some clinicians disclosed that creating opportunities for reimbursement would help address institutional barriers that limit the time available for counseling on the ERPO process would make them more willing to engage in counseling. This sentiment particularly resonated with those who had high caseloads and limited resources due to the COVID-19 pandemic.

**Voluntary removal options.** Clinicians who believed ERPOs violated their patients' Second Amendment rights or who feared involving the legal system would damage therapeutic relationships or put patient safety at risk explained that the availability of voluntary firearm removal options would alleviate this concern. Social workers who supported specific patient populations, especially youth, indicated that temporary and voluntary firearm removal options would be helpful for households where the patient was not the gun owner but had easy access to a firearm in the home.

**Ability to refer patients to someone else.** The ability to refer patients to another clinician (e.g., primary care provider or social worker), the legal system, or a designated ERPO specialist was identified across all provider types as a facilitator to address each barrier named by participants. This facilitator was proposed as an option by both those willing and those who were hesitant to counsel patients and/or their families or contact law enforcement about ERPOs. The nuances of the proposed range of alternative roles are explored in the next section.

## Alternative provider to take on role of counseling, consulting, or using ERPOs

Clinicians who were unwilling to take on the role of counseling about ERPOs or contacting law enforcement often described these actions as outside the scope of their practice. For some participants, this was because they were a subspecialty provider (e.g., pediatric ophthalmology). Many suggested other clinicians, law enforcement, or a dedicated ERPO specialist to be more appropriate to take on this role (Table 3).

**Primary care providers.** Many subspecialty clinicians highlighted the benefits of referring patients they identify as in potential need of an ERPO back to their primary care provider. They explained patients' primary care providers would have a more comprehensive picture of patients' risk and protective factors for harm, as well as the relationship needed to navigate this potentially difficult conversation.

**Psychiatrists/mental health providers.** Some clinicians, including primary care providers, preferred to refer patients to psychiatrists or other mental health providers. They explained they did not believe they had the expertise to support patients who were experiencing a behavioral health crisis severe enough to necessitate filing an ERPO. One physician explained,

> "I generally think that if patient safety concerns rise to the level of possible suicidal/homicidal actions, mental health providers need to be heavily involved in determining how to best address such concerns and provide other treatment/support to the patient. Thus, it would be most appropriate for [ERPOs] to be within the scope of psychiatrists."

Additionally, some social workers indicated that because of the structure of the medical system, that a psychologist or psychiatrist's final determination would hold more authority in a legal setting,

> "I do feel that this topic requires interdisciplinary involvement and oversight and that physicians/psychiatrists need to be a part of the process as well, as their diagnostic documentation often holds more weight in court, similarly in cases involving decisional capacity, conservatorship, and psychiatric holds, for example."

**Social workers.** Physicians and ARNPs often described counseling about or using ERPOs as the role of social workers, explaining they were better trained and had more time to evaluate and support these patients. One physician explained,

**Table 3. Suggested alternative providers to take on role of counseling or contacting law enforcement about ERPO.**

| Alternative Provider Type | Strengths | Limitations |
|---|---|---|
| Primary care providers | More comprehensive understanding of risk and protective factors and relationship to navigate conversation | Some unwilling to take on role due to previously described barriers |
| Psychiatrists and other mental health providers | Expertise to support clients in crisis severe enough to need ERPO | Some unwilling to take on role due to previously described barriers |
| Social workers | Perceived by physicians/ARNPs to have more time and training | May not align with social work foundational values and practices, especially around client self-determination |
| Law enforcement | Already involved in firearm removal and perceived by some clinicians to be in law enforcements' purview | Some clinicians concerned about patient safety and client relationship when law enforcement is involved |
| ERPO liaison | Would have specialized knowledge and dedicated time | May not be accessible depending on location and demand |

"I agree the clinic is a good setting to intervene to reduce gun violence, but it is not a physician role. The state needs to put social workers or legal resources into the clinic if this is where the intervention should take place."

There were also suggestions from social workers to utilize sub-specialty social workers to engage with ERPOs. Social workers listed existing consultation resources for referrals in Washington state (e.g., Designated Crisis Responders [DCRs]) that they found meaningful for comparable patient concerns.

"I think specialized social workers (like DCR) similar type of profession) could/should do it. But not any social worker."

**Law enforcement.** Some clinicians, especially those who expressed concerns about the constitutionality of ERPOs, believed any engagement with ERPOs was outside the scope of clinicians. A few providers believed law enforcement officers were better equipped to counsel about or seek an ERPO, with one ARNP explaining,

"In my rural county, law enforcement personnel are exceptionally skilled in de-escalation skills and demonstrating kindness to people having mental health crises."

A social worker similarly endorsed components of the legal system as a more appropriate option,

"I don't think social workers are responsible for enacting gun control legislation. That should be the domain of the legal system and government gun control policy."

**ERPO specialist.** Numerous clinicians explained that having access to an ERPO specialist would address issues related to time and staffing limitations. One physician explained,

"I think that some type of centralized/state-run firearm injury prevention support structure for all healthcare providers could have the potential to vastly improve the current state of things. Taking the work out of the process is key to success."

Additionally, one clinician explained having access to an ERPO specialist would address their concerns about the therapeutic relationship with the patient, saying,

"Having a non-medical third party as an intermediary to reduce the patient's perception that their healthcare providers have betrayed their trust."

## Discussion

In this study, we describe barriers and facilitators to the integration of ERPO counseling or connecting with law enforcement into the clinical workflow from the perspective of clinicians who were hesitant or requested additional resources to do so. This study also illustrates the strengths and limitations of alternative professionals that some clinicians in this study identified as more appropriate to take on this role.

We expand on prior research describing barriers and facilitators to clinicians' involvement in ERPOs [11, 12] in several ways. First, we conducted this research in a state where clinicians cannot independently file an ERPO, which is the case for most states with an ERPO law. Most work to date has focused on the perceived ERPO barriers and facilitators among physicians who can independently file an ERPO petition. However, preliminary analyses of ERPO petitions in a state where clinicians cannot independently file has shown clinicians can still be involved in the ERPO process by counseling patients/families about ERPOs or contacting law enforcement [13]. Our study therefore provides an initial insight into the barriers and facilitators of integrating ERPO counseling into the clinical workflow of providers who do not independently file.

Second, we explored the role of non-physician clinicians (i.e., ARNPs and social workers) in counseling patients or contacting law enforcement about ERPOs. In this study, we found similar attitudes in many of the barriers and facilitators across all clinician types. For example, concerns about knowledge gaps, lack of time, reimbursement structures, patient and clinician safety, and the potential for damaging the therapeutic relationship with patients were common for all provider types. In contrast, physicians and ARNPs more frequently identified concerns about restricting patient constitutional rights, while social workers were more focused on ERPOs potentially violating patients' self-determination through an involuntary approach to firearm removal. Additionally, while some participants across provider types described ERPOs as outside of their professional scope, the framing of this concern differed. Specifically, physicians and ARNPs described ERPOs as outside of their practice scope (i.e., practicing medicine), yet social workers often described ERPOs as contradictory to the foundational ethics of the profession. Additionally, social workers explained they may not be able to counsel patients/their families or contact law enforcement about ERPOs due to institutional policies; these concerns were not described by physicians or ARNPs. These differences signal areas where training needs and requirements may differ between clinician types. Additional clinician education about voluntary firearm removal options would address many of these barriers described across clinician types.

This study additionally extends discussion of existing barriers and facilitators by illustrating nuanced ways to address barriers to ERPO use that could be useful to improve ERPO laws. While most survey participants indicated willingness to engage with ERPOs in their clinical practice, we emphasize the reservations of and additional support requests among clinicians who indicated hesitancy toward utilizing ERPOs. Regardless of previous knowledge of ERPOs, participants were keenly aware of limitations they faced that might hinder their ability to counsel patients or contact law enforcement about ERPOs, particularly considering COVID-19's impact on resources and staffing. Therefore, the ability to refer these patients to someone else was identified as a facilitator corresponding to every barrier. Some clinicians identified a specialized role such as an ERPO specialist could potentially address the nuances of when ERPOs are appropriate, rather than supporting a blanket policy for recommending ERPOs for any patient who might be at elevated risk of firearm-related harm. This role was also identified as an opportunity to support clinicians in the ERPO process without engaging them as independent filers.

Although an ERPO specialist could address clinicians' identified barriers, including some facets of conflict with their profession, this alternative role would not meet the needs of providers who identified that the ERPO process itself was a risk to their clients' safety and autonomy. Some cited that ERPO counseling or initiation might alienate patients, not only in their care setting, but also regarding that patient's likelihood to seek medical care at all in the future. In lieu of incorporating ERPO counseling or referral into their practice setting, alternative options were recommended, focused on voluntary safe storage and/or removal (e.g., free gun

safes, firearm locking devices, temporary or long-term firearm storage locations) [17–22]. Clinicians who offered these examples of voluntary removal, or requested information on voluntary options, cited the need to respect their clients' autonomy and/or safety from the legal system. Expanding provider awareness of and access to voluntary options was also included as a resource that could be used when safety planning before acute risk of firearm injury or death could arise.

Clinicians described a rich array of perceived barriers and facilitators to ERPOs that considered complex needs their patients might have when experiencing a behavioral health crisis, such as suicidal ideation, homicidal ideation, or intimate partner violence. These concerns highlight need for additional training in several areas. Some clinicians showed bias in assuming clients with mental illness are more likely to be violent; trainings should include research demonstrating this inaccuracy which perpetuates stigma [23, 24]. Additionally, many clinicians who described counseling about ERPOs as outside of their practice scope also expressed a broader discomfort with assessing for suicidal or homicidal ideation, indicating another domain for future training. Finally, in addition to the need for research on the effectiveness of ERPOs, efforts are needed to ensure research is translated and communicated to clinicians promptly.

## Limitations

Limitations of this study include the low response rate and potential response bias of survey participants. Because firearm policies are polarizing, it is possible some recipients assumed the survey was designed to collect data as evidence to restrict firearm access, and so were less likely to complete it. It is also likely that those who responded to the open-ended questions analyzed in this study had stronger opinions about ERPOs and/or firearms than those who did not complete these questions. As such, the results of this study should not be considered generalizable to clinicians in Washington state or nationally. We were also unable to account for the political views of participants or the potential impact of the COVID-19 pandemic on survey fatigue or participants' views. Our survey focused on the barriers to engaging with ERPOs in a clinical setting and thus we have provided a nuanced explanation of the concerns some clinicians have around their role in ERPOs and highlight their rationale for other professionals they suggest are better equipped to take on this role. Finally, we were unable to obtain contact information for psychologists or other mental health providers, who also serve clients at high risk of harming themselves or someone else with a firearm.

## Conclusions

Clinician perspectives on incorporating ERPOs into their clinical workflow ranged widely within each provider type and across the three groups. These differentiations were largely due to provider practice settings, the populations they serve, and moral/ethical beliefs about firearms. Preliminary findings suggest that standardized screeners, protocols, or deputizing all clinicians to be able to independently file ERPOs may not be appropriate due to the complexity of firearm ownership across geographic locations and populations. This study highlights the need for additional research to identify implementation needs for providers in clinical settings where ERPOs may be relevant. Training areas include education on counseling about voluntary firearm removal options, best practices on assessing patients for suicidal or homicidal ideation, and interventions to reduce access to lethal means (e.g., ERPOs).

## Supporting information

**S1 Dataset.**
(XLSX)

## Acknowledgments

The authors thank Julia Schleimer for her consultation on this project.

## Author Contributions

**Conceptualization:** Kelsey M. Conrick, Sarah F. Porter, Emma Gause, Ali Rowhani-Rahbar, Frederick P. Rivara, Megan Moore.

**Data curation:** Sarah F. Porter, Emma Gause.

**Formal analysis:** Kelsey M. Conrick, Sarah F. Porter, Laura Prater.

**Funding acquisition:** Kelsey M. Conrick, Ali Rowhani-Rahbar, Frederick P. Rivara.

**Investigation:** Kelsey M. Conrick, Sarah F. Porter.

**Methodology:** Kelsey M. Conrick, Emma Gause.

**Resources:** Megan Moore.

**Supervision:** Ali Rowhani-Rahbar, Frederick P. Rivara, Megan Moore.

**Visualization:** Kelsey M. Conrick, Sarah F. Porter.

**Writing – original draft:** Kelsey M. Conrick, Sarah F. Porter.

**Writing – review & editing:** Emma Gause, Laura Prater, Ali Rowhani-Rahbar, Frederick P. Rivara, Megan Moore.

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
