## [Decision Letter · Decision Letter 0]

19 May 2023

PONE-D-23-09742

Integration of extreme risk protection orders into the clinical workflow: qualitative comparison of clinician perspectives

PLOS ONE

Dear Dr. Conrick,

Thank you for submitting your manuscript to PLOS ONE. After careful consideration, we feel that it has merit but does not fully meet PLOS ONE’s publication criteria as it currently stands. Therefore, we invite you to submit a revised version of the manuscript that addresses the points raised during the review process.

Overall, the reviewers and I thought your article was very well done and a timely addition to knowledge on implementation of ERPOs. There are a few things that you should focus on in revision. First, think about how some of your examples tend to present a perception that most clinicians would not want to be involved in ERPOs, which I think is not your finding, but how the paper reads. Second, please address whether your sample excluded or included mental health clinicians per the query from reviewer 2. This then leads to a need to comment on the acceptance or lack thereof by mental health clinicians compared to other disciplines.

We look forward to receiving your revised manuscript.

Kind regards,

James Curtis West, M.D.

Academic Editor

PLOS ONE

“This publication was supported by the National Center For Advancing Translational Sciences of the National Institutes of Health under Award Number TL1TR002318 (to KMC). The content is solely the responsibility of the authors and does not necessarily represent the official views of the National Institutes of Health. This study was supported by funds from the State of Washington to the Firearm Injury & Policy Research Program.”

Reviewers' comments:

Reviewer's Responses to Questions

**Comments to the Author**

1. Is the manuscript technically sound, and do the data support the conclusions?

Reviewer #1: Yes

Reviewer #2: Yes

2. Has the statistical analysis been performed appropriately and rigorously? 

Reviewer #1: N/A

Reviewer #2: N/A

3. Have the authors made all data underlying the findings in their manuscript fully available?

Reviewer #1: Yes

Reviewer #2: Yes

4. Is the manuscript presented in an intelligible fashion and written in standard English?

Reviewer #1: Yes

Reviewer #2: Yes

5. Review Comments to the Author

Reviewer #1: Thank you for the opportunity to review this manuscript, in which the authors conducted a large scale study of clinician attitudes towards ERPO in WA. Overall, this is a very helpful analysis of perceived barriers and potential facilitators. My comments are very superficial.

To the unfamiliar reader, the paper overall gives the impression that WA clinicians are generally against ERPO, listing many problems with the laws. In fact, as this study teams’ other papers have shown, the majority of WA clinicians are very willing to file ERPOs or to work with other parties to file for them (Gause et al 2022 in Prev Med). I would suggest dedicating another sentence or two in discussion to the fact that despite these perceived barriers, MOST clinicians do want to engage in ERPO (the authors can cite themselves here). These facilitators would add to this, but even without them in place, these ERPOs are clearly desired tools.

In the Conclusions section, the authors mention that responses varied across practice setting etc. However, although they referenced differences in provider type regularly, practice setting does not seem to come up in distinguishing responses across barrier/facilitator domains. Maybe it would be helpful to provide a table that not only characterized the respondents by provider type (As is done in text) but by practice setting and anything else appropriate for an opening table (years of experience, sex, whatever else was captured in the survey) and then refer to practice setting among these other things throughout results in order to illustrate any differences in attitudes that may exist. This is not an essential, but would help readers make the most sense of the findings.

Very minor points:

Line 55 (“more than 90%”): The cited study found ‘89.6%’ fatality rate, so it would be more accurate to just say “90%” or “about 90%” rather than “more than 90%”.

One quote is used twice. With thousands of respondents, it shouldn’t be too hard to find a different quote to use for one of the examples.

“We have a constitutional right to keep and bear arms, though public and

personal safety are also key. Mentally ill people already carry so much stigma I

wonder if this would help or make it worse? Also, no contact orders frequently

fail to prevent contact and there are other means of obtaining firearms than

through legitimate sources so would an ERPO really be effective?”

Overall, this is a great paper, and it adds to our understanding of barriers to ERPOs and potential ways to circumnavigate them.

Reviewer #2: This study used a qualitative approach to examine the clinical perspectives of Extreme Risk Protection Orders (ERPO) in clinical care in Washington State. The sample consisted of 4,242 license physicians, nurse practitioners, and social workers in Washington State who responded to a survey distributed between May – July, 2021. There is a paucity of research on the effects of ERPOs, which may inform the development of interventions to prevent firearm-related injury and death. A main limitation in the study is the results are not generalizable to clinicians in other states where ERPOs laws may differ, and psychologists and other mental health providers were not included in the sample; however, this study fills a research gap, is very timely, and an important topic.

The following major revision may improve the manuscript:

Abstract:

Page 2, Line 32. Please mention how the survey was conducted either online or by mail.

Introduction:

Page 3, Line 49. Please define the term firearm-related harm prior to using throughout the manuscript. Harm is a broad term that can refer to self-harm, suicidal behaviors, injuries, death, ect…

Page 4, Line 72. Is this true for other states? This reference only refers to Washington State. Additionally, screening for access to lethal means is standard procedure in military populations.

Page 4, Lines 86, 87 This sentence is a repeat from above.

Page 4, Line 87. Clearly state the objective of the study here, which is to better understand the barriers to ERPOs use and the utilization of ERPOs in Washington State as reported by physicians, social workers and nurse practitioners.

Methods

Design Overview:

Page 5, Line 96. Please include more detail of the original study in addition to the citation of the broader study, in this manuscript.

Participants and Procedures:

Page 5, Line 104. Please explain the rationale for the exclusion of psychologists and other mental health providers, who also serve those at highest risk of firearm related suicide.

Page 6, Line 122. The response rates for physicians (25%) ARNPs (24%) and social workers (30%) are low and need to be addressed in the limitations. How does non-response bias the findings?

Results

Page 7, Line 147. Please explain what this statement means … “ of who should do so.”

Page 7, Lines 149-153. Fig 1. Conceptual Model of Findings:

Please format the lines 150-153. It may be clearer to the reader to state how each facilitator addresses each barrier rather than stating “ facilitators to the barrier ”

Page 10, Lines 192-193. It will be important to emphasize the need for more research on ERPOs effectiveness in the future.

Page 11, Lines 197-199. Please explain this statement and how ERPOs are related to this specific situation?

Discussion:

Please mention these additional limitations: the political views of the participants could not be controlled for in descriptive qualitative analyses, and the effects of the pandemic were not assessed. The low response rate and response bias also need to be addressed.

Tables:

Please format the tables according to the journal requirements. Please add the n for each table in the title. Footnotes should be single-spaced and in a smaller font.

Table 1, 2 and Fig. 1: n should be added and italicized

6. PLOS authors have the option to publish the peer review history of their article (what does this mean?). If published, this will include your full peer review and any attached files.

Reviewer #1: No

Reviewer #2: No

---

## [Author Response · Author response to Decision Letter 0]

16 Jun 2023

Journal requirements

Journal Requirement Comment 1: Please ensure that your manuscript meets PLOS ONE's style requirements, including those for file naming. The PLOS ONE style templates can be found at https://journals.plos.org/plosone/s/file?id=wjVg/PLOSOne_formatting_sample_main_body.pdf and https://journals.plos.org/plosone/s/file?id=ba62/PLOSOne_formatting_sample_title_authors_affiliations.pdf

 Author response: We have updated our file formatting as requested.

Journal Requirement Comment 2: Thank you for stating the following financial disclosure:

“This publication was supported by the National Center For Advancing Translational Sciences of the National Institutes of Health under Award Number TL1TR002318 (to KMC). The content is solely the responsibility of the authors and does not necessarily represent the official views of the National Institutes of Health. This study was supported by funds from the State of Washington to the Firearm Injury & Policy Research Program.”

Author response: We have added a statement indicating the funder had no role in the study into the cover letter.

Journal Requirement Comment 3: We note that you have indicated that data from this study are available upon request. PLOS only allows data to be available upon request if there are legal or ethical restrictions on sharing data publicly. For more information on unacceptable data access restrictions, please see http://journals.plos.org/plosone/s/data-availability#loc-unacceptable-data-access-restrictions.

Author response: We have uploaded the anonymized data set as supporting information, and indicated this in our cover letter.

Journal Requirement Comment 4: Please include your full ethics statement in the ‘Methods’ section of your manuscript file. In your statement, please include the full name of the IRB or ethics committee who approved or waived your study, as well as whether or not you obtained informed written or verbal consent. If consent was waived for your study, please include this information in your statement as well.

Author Response: We have condensed our full ethics statement into the first paragraph of the Methods section. On page 3, lines 94-100: 

“The University of Washington Institutional Review Board (IRB) approved all study procedures. Formal consent was not obtained as the IRB deemed the research human subjects research that qualifies for exempt status; participants were informed of the nature of the survey, reminded that participation was voluntary and could be revoked at any time, and provided with contact information for the research team should they have questions or concerns. The research team had access to information that could identify individual participants; these data were used for recruitment purposes only and were not accessed during analysis.”

Journal Requirement Comment 5: Please review your reference list to ensure that it is complete and correct. If you have cited papers that have been retracted, please include the rationale for doing so in the manuscript text, or remove these references and replace them with relevant current references. Any changes to the reference list should be mentioned in the rebuttal letter that accompanies your revised manuscript. If you need to cite a retracted article, indicate the article’s retracted status in the References list and also include a citation and full reference for the retraction notice.

 Author response: We have reviewed the reference list and confirmed its accuracy. 

 

Reviewer's Responses to Questions

Reviewer 1

Reviewer Comment 1: Thank you for the opportunity to review this manuscript, in which the authors conducted a large scale study of clinician attitudes towards ERPO in WA. Overall, this is a very helpful analysis of perceived barriers and potential facilitators. My comments are very superficial.

Author Response: The authors thank the reviewer for their detailed comments to improve the scientific rigor and communication of findings for this manuscript. 

Reviewer Comment 2: To the unfamiliar reader, the paper overall gives the impression that WA clinicians are generally against ERPO, listing many problems with the laws. In fact, as this study teams’ other papers have shown, the majority of WA clinicians are very willing to file ERPOs or to work with other parties to file for them (Gause et al 2022 in Prev Med). I would suggest dedicating another sentence or two in discussion to the fact that despite these perceived barriers, MOST clinicians do want to engage in ERPO (the authors can cite themselves here). These facilitators would add to this, but even without them in place, these ERPOs are clearly desired tools.

Author Response: We agree our initial framing of the participants who were included in our qualitative analysis was unclear and may have overrepresented the proportion of clinicians who expressed concerns about the barriers and ethical concerns about ERPOs. We have made the following edits to clarify this point.

On page 3, lines 72-73: “Most clinicians have expressed willingness to independently file ERPOs; however, some clinicians remain hesitant due to lack of knowledge, concerns about time, and fears of liability [10–13].”

On page 3, lines 83-89: “Despite the potential benefits of including clinicians in the process, some have expressed hesitancy to counsel patients and their families about ERPOs or engage with law enforcement. Reasons for these clinicians’ hesitancy include confusion about the process, lack of time, liability concerns, and personal safety concerns [10,11,13]. We sought to better understand the concerns of physicians, nurse practitioners, and social workers who expressed hesitancy to be involved in the ERPO process in Washington State, where clinicians cannot independently file ERPOs. Specifically, we considered their concerns about the integration of ERPOs into their clinical workflow.”

On page 4, line 123-126: “We used content analysis to identify themes and categories related to barriers and facilitators to clinicians’ willingness to counsel patients/their families or contact law enforcement about ERPOs among those who expressed hesitancy or requested additional resources to do so [15].”

On page 13, lines 339-341: “In this study, we describe barriers and facilitators to the integration of ERPO counseling or connecting with law enforcement into the clinical workflow from the perspective of clinicians who were hesitant or requested additional resources to do so.”

On page 13, lines 374-376: “While most survey participants indicated willingness to engage with ERPOs in their clinical practice, we emphasize the reservations of and additional support requests among clinicians who indicated hesitancy toward utilizing ERPOs.” 

Reviewer Comment 3: In the Conclusions section, the authors mention that responses varied across practice setting etc. However, although they referenced differences in provider type regularly, practice setting does not seem to come up in distinguishing responses across barrier/facilitator domains. Maybe it would be helpful to provide a table that not only characterized the respondents by provider type (As is done in text) but by practice setting and anything else appropriate for an opening table (years of experience, sex, whatever else was captured in the survey) and then refer to practice setting among these other things throughout results in order to illustrate any differences in attitudes that may exist. This is not an essential, but would help readers make the most sense of the findings.

Author response: We agree with the reviewer that the practice setting of clinicians and their background information are important to understanding the context of their concerns and additional resources needed. We do not provide a table of practice settings because the answer options were significantly different between the surveys for each provider type. Instead, we highlight the influence of clinicians’ practice settings where they identify that context as relevant to their concern. We have clarified the language and influence of practice setting in the following places:

On pages 4-5, lines 136-140: “Hesitation or unwillingness to engage with ERPOs for some clinicians was dependent on their practice setting and communities they served. Common concerns included whether ERPOs fell into their professional scope, knowledge gaps, institutional barriers, perceived ERPO effectiveness and constitutionality, and potential implications ERPOs may have on their patients, their patient’s families, or themselves.”

On page 7, lines 181-185: “All three provider types cited contextual factors unique to their practice setting that created barriers or concerns to ERPO implementation. In addition to the unique environmental considerations within and across practice settings, particularly between rural versus urban locations, common institutional barriers included staffing shortages, reimbursement structures, time, and perceived lack of institutional support for ERPOs.”

Very minor points:

Reviewer Comment 4: Line 55 (“more than 90%”): The cited study found ‘89.6%’ fatality rate, so it would be more accurate to just say “90%” or “about 90%” rather than “more than 90%”.

 Author response: We have corrected this to say “about 90%.”

On page 2, lines 54-56: “As firearms have about a 90% case-fatality rate for self-inflicted injury [4], several policies have been enacted seeking to reduce access to firearms for individuals acutely at risk of harming themselves and/or others.”

Reviewer Comment 5: One quote is used twice. With thousands of respondents, it shouldn’t be too hard to find a different quote to use for one of the examples. “We have a constitutional right to keep and bear arms, though public and personal safety are also key. Mentally ill people already carry so much stigma I wonder if this would help or make it worse? Also, no contact orders frequently fail to prevent contact and there are other means of obtaining firearms than

through legitimate sources so would an ERPO really be effective?”

Author response: We appreciate the reviewer’s identification of this duplication and have found a different quote to highlight perceived ERPO effectiveness. 

In Table 1: “Often I have clients, especially young people, say 'I don't have a gun but I can easily get one from somebody' which these tools would not capture this population. Not an easy fix, but just wanted to point this out as I hear it quite often and it always worries me as it is hard to make a safe plan around potential to obtain from other people.”

 

Reviewer 2

Reviewer Comment 1: This study used a qualitative approach to examine the clinical perspectives of Extreme Risk Protection Orders (ERPO) in clinical care in Washington State. The sample consisted of 4,242 license physicians, nurse practitioners, and social workers in Washington State who responded to a survey distributed between May – July, 2021. There is a paucity of research on the effects of ERPOs, which may inform the development of interventions to prevent firearm-related injury and death. A main limitation in the study is the results are not generalizable to clinicians in other states where ERPOs laws may differ, and psychologists and other mental health providers were not included in the sample; however, this study fills a research gap, is very timely, and an important topic.

Author response: The authors thank the reviewer for their detailed comments to improve the scientific rigor and communication of findings for this manuscript. We agree our findings may not be generalizable to states where clinicians can independently file ERPOs. However, given most states with an ERPO law do not let clinicians file, we do believe our findings may be applicable to other states. More research is needed to confirm our findings. Additionally, we agree the inclusion of psychologists and other mental health providers would have strengthened our study. We unfortunately were unable to obtain the contact information for these providers. 

The following major revision may improve the manuscript:

Reviewer Comment 2: 

Abstract:

Page 2, Line 32. Please mention how the survey was conducted either online or by mail.

Author response: We have clarified the survey was conducted online.

On page 2, lines 32-34: “We analyzed responses to open-ended questions from an online survey distributed May-July of 2021 to all licensed physicians (n=23,051), nurse practitioners (n=8,049), and social workers (n=6,910) in Washington state.”

On page 3, lines 91-94: “We analyzed responses to ten open-ended questions (Appendix A) from an online survey conducted May-June 2021 with clinicians in Washington state regarding their opinions of ERPOs.”

Reviewer Comment 3: 

Introduction:

Page 3, Line 49. Please define the term firearm-related harm prior to using throughout the manuscript. Harm is a broad term that can refer to self-harm, suicidal behaviors, injuries, death, ect…

Author response: We agree with the reviewer that our language was unclear. We have clarified the opening paragraph to focus only on firearm injuries and deaths related to interpersonal and self-directed harm. While other types of firearm-related harm, such as intimidation, direct threats, use in crime, or community exposure to firearm violence are critical, this study focuses on the role of clinicians in preventing interpersonal (harm to others) and self-directed (harm to self) injuries and deaths. 

On page 2, lines 50-54: “The United States (US) faces a unique and worsening burden of firearm-related injuries and deaths. In 2021, nearly 49,000 individuals died from firearm injuries [1], and an estimated 86,000 were injured [3]. Firearms were used in 26,320 suicides in the US during 2021—the highest since 1993 [2]. Lethal means restriction policies seek to prevent injury and death by reducing access to the most lethal means of harm.”

Reviewer Comment 4:

Page 4, Line 72. Is this true for other states? This reference only refers to Washington State. Additionally, screening for access to lethal means is standard procedure in military populations.

Author response: We have updated the citation to reference a scoping review of 53 articles that found low rates of clinician screening for access to firearms across patient populations and practice settings. We have also clarified the language to indicate we are referring to assessing access to firearms specifically, rather than lethal means broadly.

On page 3, lines 70-71: “Clinicians’ assessment of access to lethal means can potentially reduce suicide risk in high-risk patients [7]; however, very few providers screen for access to firearms regularly [8].”

[8] Roszko PJD, Ameli J, Carter PM, Cunningham RM, Ranney ML. Clinician Attitudes, Screening Practices, and Interventions to Reduce Firearm-Related Injury. Epidemiologic Reviews. 2016;38: 87–110. doi:10.1093/epirev/mxv005

Reviewer Comment 5:

Page 4, Lines 86, 87 This sentence is a repeat from above.

 Author response: We have removed this duplicate sentence.

Reviewer Comment 6:

Page 4, Line 87. Clearly state the objective of the study here, which is to better understand the barriers to ERPOs use and the utilization of ERPOs in Washington State as reported by physicians, social workers and nurse practitioners.

Author response: We have reworded the purpose statement according to both reviewers’ suggestions. It now reads:

On page 3, lines 85-89: “We sought to better understand the concerns of physicians, nurse practitioners, and social workers who expressed hesitancy to be involved in the ERPO process in Washington State, where clinicians cannot independently file ERPOs. Specifically, we considered their concerns about the integration of ERPOs into their clinical workflow.”

Methods

Reviewer Comment 7:

Design Overview:

Page 5, Line 96. Please include more detail of the original study in addition to the citation of the broader study, in this manuscript.

Author response: We have removed this sentence from the design overview, as the fielding and survey instrument are described in subsequent sections. We have added a sentence referring to the two publications that describe the quantitative findings from the survey. 

On page 4, lines 116-117: “Results for the quantitative components of the survey are available in other publications [9,10].”

Participants and Procedures:

Reviewer Comment 8:

Page 5, Line 104. Please explain the rationale for the exclusion of psychologists and other mental health providers, who also serve those at highest risk of firearm related suicide.

Author response: We agree the inclusion of psychologists and other mental health providers would have strengthened our study. We unfortunately were unable to obtain the contact information for these providers. We have explained this in the methods section, as well as added it to the limitations section.

On page 4, lines 105-106: “We were unable to obtain contact information for psychologists or other mental health providers.”

On page 15, lines 421-423: “Finally, we were unable to obtain contact information for psychologists or other mental health providers, who also serve clients at high risk of harming themselves or someone else with a firearm.”

Reviewer Comment 9:

Page 6, Line 122. The response rates for physicians (25%) ARNPs (24%) and social workers (30%) are low and need to be addressed in the limitations. How does non-response bias the findings?

Author response: These percentages refer to the proportion of participants who responded to at least one open-ended question of all survey participants. These numbers are low in part due to the fact that many open-ended questions relied on branching logic. That is, a participant would have to select “other” or “I don’t think clinicians should counsel about ERPOs” to be prompted with an open-ended response box. We have listed the questions and branching logic on the data file submitted with the manuscript. We additionally explain the effect of response bias both to the survey and to the open-ended questions on our findings in the limitations.

On pages 14-15, lines 410-421: “Limitations of this study include the low response rate and potential response bias of survey participants. Because firearm policies are polarizing, it is possible some recipients assumed the survey was designed to collect data as evidence to restrict firearm access, and so were less likely to complete it. It is also likely that those who responded to the open-ended questions analyzed in this study had stronger opinions about ERPOs and/or firearms than those who did not complete these questions. As such, the results of this study should not be considered generalizable to clinicians in Washington state or nationally. We were also unable to account for the political views of participants or the potential impact of the COVID-19 pandemic on survey fatigue or participants’ views. Our survey focused on the barriers to engaging with ERPOs in a clinical setting and thus we have provided a nuanced explanation of the concerns some clinicians have around their role in ERPOs and highlight their rationale for other professionals they suggest are better equipped to take on this role.”

Results

Reviewer Comment 10:

Page 7, Line 147. Please explain what this statement means … “ of who should do so.”

Author response: We have clarified that the sentence refers to clinicians’ opinions of who should counsel or contact law enforcement about ERPOs, among those who did not wish to counsel/contact law enforcement themselves.

On page 5, lines 145-148: “Among those who were unwilling or hesitant to engage with ERPOs in their current capacity (counsel patients/their families or contact law enforcement), these barriers and facilitators influenced their perspective of who should counsel or contact law enforcement about ERPOs.”

Reviewer Comment 11:

Page 7, Lines 149-153. Fig 1. Conceptual Model of Findings:

Please format the lines 150-153. It may be clearer to the reader to state how each facilitator addresses each barrier rather than stating “ facilitators to the barrier ”

Author response: We have updated the language in this caption.

On page 5, lines 153-156: “The conceptual model describes which clinician-identified barriers are addressed by each facilitator to Extreme Risk Protection Order (ERPO) counseling integration into the clinical workflow. For example, trainings about ERPOs and the ability to refer patients to someone else were identified as facilitators to address the barrier of knowledge gaps.”

Reviewer Comment 12:

Page 10, Lines 192-193. It will be important to emphasize the need for more research on ERPOs effectiveness in the future.

Author response: We have added this as a call for future research in the discussion section.

On page 14, lines 407-408: “Finally, in addition to the need for research on the effectiveness of ERPOs, efforts are needed to ensure research is translated and communicated to clinicians promptly.”

Reviewer Comment 13:

Page 11, Lines 197-199. Please explain this statement and how ERPOs are related to this specific situation?

Author response: We have clarified this language to indicate we are referring to patients who reported being able to access firearms despite an ERPO.

On page 7, lines 204-207 “Clinicians expressed doubts about the effectiveness of ERPOs for specific patients, citing instances where patients disclosed their ability to obtain firearms through alternative channels, despite confiscation of presently owned firearms and prohibition of purchase as a result of the ERPO.”

Discussion:

Reviewer Comment 14:

Please mention these additional limitations: the political views of the participants could not be controlled for in descriptive qualitative analyses, and the effects of the pandemic were not assessed. The low response rate and response bias also need to be addressed.

Author response: We have updated the limitations section. It now reads:

On pages 14-15, lines 410-421: “Limitations of this study include the low response rate and potential response bias of survey participants. Because firearm policies are polarizing, it is possible some recipients assumed the survey was designed to collect data as evidence to restrict firearm access, and so were less likely to complete it. It is also likely that those who responded to the open-ended questions analyzed in this study had stronger opinions about ERPOs and/or firearms than those who did not complete these questions. As such, the results of this study should not be considered generalizable to clinicians in Washington state or nationally. We were also unable to account for the political views of participants or the potential impact of the COVID-19 pandemic on survey fatigue or participants’ views. Our survey focused on the barriers to engaging with ERPOs in a clinical setting and thus we have provided a nuanced explanation of the concerns some clinicians have around their role in ERPOs and highlight their rationale for other professionals they suggest are better equipped to take on this role.”

Tables:

Reviewer Comment 15:

Please format the tables according to the journal requirements. Please add the n for each table in the title. Footnotes should be single-spaced and in a smaller font.

Table 1, 2 and Fig. 1: n should be added and italicized

Author response: We have corrected the tables to match journal requirements, including adding the sample size to the title of the table. We do not have footnotes in any tables.

---

## [Editor Report · Decision Letter 1]

6 Jul 2023

Integration of extreme risk protection orders into the clinical workflow: qualitative comparison of clinician perspectives

PONE-D-23-09742R1

Dear Dr. Conrick,

We’re pleased to inform you that your manuscript has been judged scientifically suitable for publication and will be formally accepted for publication once it meets all outstanding technical requirements.

Kind regards,

James Curtis West, M.D.

Academic Editor

PLOS ONE
---

## [Editor Report · Acceptance letter]

15 Aug 2023

PONE-D-23-09742R1 

Integration of extreme risk protection orders into the clinical workflow: qualitative comparison of clinician perspectives 

Dear Dr. Conrick:

I'm pleased to inform you that your manuscript has been deemed suitable for publication in PLOS ONE. Congratulations! Your manuscript is now with our production department. 

Kind regards, 

on behalf of

Dr. James Curtis West 

Academic Editor

PLOS ONE